# Multiscale Theoretical Study of Sulfur Dioxide (SO_2_) Adsorption in Metal–Organic Frameworks

**DOI:** 10.3390/molecules28073122

**Published:** 2023-03-31

**Authors:** Charalampos G. Livas, Dionysios Raptis, Emmanuel Tylianakis, George E. Froudakis

**Affiliations:** 1Department of Chemistry, University of Crete, Voutes Campus, GR-71003 Heraklion, Crete, Greece; 2Department of Materials Science and Technology, University of Crete, Voutes Campus, GR-71003 Heraklion, Crete, Greece

**Keywords:** Metal–Organic Frameworks (MOFs), sulfur dioxide (SO_2_), Density Functional Theory (DFT), Grand Canonical Monte Carlo (GCMC)

## Abstract

In the present work, we used DFT in order to study the interaction of SO_2_ with 41 strategically functionalized benzenes that can be incorporated in MOF linkers. The interaction energy of phenyl phosphonic acid (–PO_3_H_2_) with SO_2_ was determined to be the strongest (−10.1 kcal/mol), which is about 2.5 times greater than the binding energy with unfunctionalized benzene (−4.1 kcal/mol). To better understand the nature of SO_2_ interactions with functionalized benzenes, electron redistribution density maps of the relevant complexes with SO_2_ were created. In addition, three of the top performing functional groups were selected (–PO_3_H_2_, –CNH_2_NOH, –OSO_3_H) to modify the IRMOF-8 organic linker and calculate its SO_2_ adsorption capacity with Grand Canonical Monte Carlo (GCMC) simulations. Our results showed a great increase in the absolute volumetric uptake at low pressures, indicating that the suggested functionalization technique can be used to enhance the SO_2_ uptake capability not only in MOFs but in a variety of porous materials.

## 1. Introduction

In the last decades, energy demands have been on the rise as the availability, cost, and consumption of energy determine any nation’s degree of social and economic development. Crude oil is still the number one source in the energy mix, surpassing natural gas, nuclear, and renewables [1]. Its production is constantly rising, a trend that was halted in 2020 when the pandemic’s effects on daily life caused a decrease in energy needs [1]. Significant quantities of sulfur oxides (SO_x_) are released into the environment after the combustion of fuels produced from petroleum in internal combustion engines used in motor vehicles. Notably, SO_2_ is the compound of great concern accounting for the vast majority of SO_x_ in the atmosphere.

SO_2_ is a colorless gas with a sultry odor and water-soluble gas. It has negative effects on the environment as it constitutes an important contributor to air pollution. According to the World Health Organization (WHO), the latter caused 4.2 million deaths in 2016. In addition, SO_2_ causes severe burns and other manifestations of eye, nose, throat, trachea, and large bronchial irritations [2]. For this reason, the creation of a desulfurization system will be a great solution for the adsorption of SO_2._ Many technologies have been proposed to control the emission of this gas [3,4]. One method used is the conversion of SO_2_ to a valuable chemical of NaHSO_4_ by means of an electrochemical reaction in an aqueous solution. This model incorporates two stages: (I) the absorption of SO_2_ gas by an aqueous solution followed by its oxidation to SO_4_(^−2^) by air and (II) the transformation of SO_4_(^−2^) to NaHSO_4_ [5]. Nevertheless, this is an energy demanding procedure, and alternative methods need to be used.

In recent years, SO_2_ adsorption from nanoporous materials has attracted scientific interest. A series of publications have been referred to. They show the experimental uptake and theoretical perdition of gas uptake from nanomaterials. Fu et al. synthesized a novel framework (CTF–CSU41) and reported that this sodium carboxylate anchored framework exhibits an exceptionally high uptake of SO_2_ with a maximum capacity of 6.7 mmol∙g^−1^ (42.9 wt%) even at low SO_2_ partial pressure of 0.15 bar (298 K). This constitutes the highest value recorded for a scrubbing material [6]. Brandt et al. [7], in an experimental work, studied the SO_2_ adsorption on MOF-177, NH2-MIL-125(Ti), and MIL-160. Especially for MOF-177, they found an exceptional uptake of 25.7 mmol/g at 293 K and 1 bar. Brandt et al. [8] measured the uptake for four Zr-MOFs and eight Al-MOFs at 293K and 1 bar and found it to be between 4.8 and 17.3 mmol/g. Lopez-Olvera et al. [9] synthesized and studied MIL-53(Al)-BDC and MIL-53(Al)-TDC for SO_2_ adsorption and found it to be 10.8 and 8.9 mmol/g, respectively, at 298 K and 1 bar. Jansen et al. [10] studied the potential of CAU-23 to adsorb various gases, including SO_2_. They found that it can adsorb 8.4 mmol g^−1^ at 293 K and 1 bar. Another study by Liusheng et al. [11] which investigated the adsorption and separation properties of SO_2_ from flue gases (SO_2_–CO_2_–N_2_) by Covalent Organic Frameworks (COF) found that the existence of an appropriate functional group and the small channels are more conducive to gas adsorption.

Yaghi originally introduced MOFs in 1995 [12,13]. Metal–Organic Frameworks are framework materials that consist of different kinds of building blocks. Metallic clusters are connected with organic linkers following different topologies, ending up in millions of possible combinations. The great variety of metallic clusters, organic linkers, and topologies that can be combined make them ideal materials for many different applications in science and technology. They have exceptional physical properties, such as extraordinarily high porosity (up to 90% free volume). Their low densities and large surface areas are distinctive characteristics which, in combination with the aforementioned feature, constitute them ideal candidates for potential uses as gas storage media and high-capacity adsorbents. The diversity of both inorganic and organic molecules building their structures plays an important role in the widespread study of different fields of these materials [14,15].

In this work, we use a multiscale approach in order to find optimum MOFs for SO_2_ storage. First, we use DFT methods to study the interaction of SO_2_ with a variety of functionalized organic linkers. In this way, our aim is to enhance this weak interaction. Then, we use Grand Canonical Monte Carlo to obtain the isotherms of SO_2_ adsorption in strategically functionalized MOFs. In this way, we can both test if the enhanced interaction is enough to produce higher adsorption and, in addition, obtain the theoretical value of SO_2_ absorption in different functionalized MOFs.

This study’s main focus is on the functionalization of organic linkers for enhancing the interactions of organic linkers with SO_2_ and thus increasing the adsorption capacity of the MOF. Since the majority of MOFs have aromatic backbones such as benzene, naphthalene, etc., as their organic linkers, we chose the simple benzene for functionalization and investigation of the interaction with SO_2_. A total of 41 functional groups were strategically selected based on both our chemical intuition and on results from previous studies conducted by our group [11,12,13] where a significant increase in the adsorption capacity of MOF materials in oxygenated gases was observed.

We investigated the nature of the interaction of SO_2_ with modified IRMOF-8 and predicted the uptake of the functionalized MOFs using both precise quantum chemistry calculations and large-scale classical Monte Carlo simulations. In addition, Quantum Mechanics data were employed to fine-tune the parameters of the force field utilized in the Grand Canonical Monte Carlo (GCMC) simulations. The latter was used to estimate adsorption isotherms at 298 K and pressure of 3 bar in order to quantify the additional storage capacity induced by the enhanced interaction of SO_2_ with the functional groups (FGs).

Our group’s prior research on other gases has demonstrated that this strategy yields notable benefits [16,17,18,19,20,21]. For example, Raptis et al. [20] revealed that, compared to the unfunctionalized one, the incorporation of functional groups leads to a 170% enhancement in the interaction energy between the functionalized benzene ring and nitrogen dioxide. A 16-fold improvement in gravimetric adsorption (mmol·g^−1^) at 1.2 bar and 298 K resulted from this increase. Additionally, Frysali et al. [18] clarified that the phenyl ring’s interaction with carbon dioxide practically doubles when a sulfate anion is present (22.6 kJ/mol).

## 2. Results and Discussion

The interaction of the SO_2_ gas molecule with 41 strategically functionalized benzenes was systematically examined and presented in Appendix A of the supporting information together with the optimized structures. Figure 1 and Figure 2 depict the optimized geometries and the binding energies respectively, of the top 11 functionalized benzenes and the unsubstituted benzene for reference. All calculations were done at the RI-DSD-BLYP/def2-TZVPP level of theory. The binding energies, in units of kcal/mol, and the percentage enhancement of the interaction compared to the simple benzene ring, for all complexes, are shown in Appendix A. The unsubstituted benzene ring exhibits an interaction energy of −4.1 kcal/mol with SO_2_, whereas the binding energies for the functionalized monomers vary from −3.5 kcal/mol to −10.1 kcal/mol. Only 5 (-NCO, -NCS, -NC, -PH_2_, -F) out of the total 41 functional groups had a negative effect and lowered the interaction with SO_2_. In contrast, 36 of them positively impacted the binding energy. This represents an initial and positive indication that the functionalization methodology promotes the interaction between the gas molecule and the substituted benzene ring.

The interaction energy between SO_2_ and phenyl phosphonic acid (–PO_3_H_2_) was found to be the highest at −10.1 kcal/mol. This energy is almost 2.5 times greater than the binding energy between SO_2_ and the unfunctionalized benzene, which is just over −4.0 kcal/mol. On the other hand, C_6_H_5_F was the case with the weakest interaction, with a binding energy of −3.5 kcal/mol. At the complex of C_6_H_5_PO_3_H_2_, the hydrogen atoms of the gas molecule are oriented toward the oxygen atoms of the substituted benzene ring and have a distance of 1.901 Å. The position of the oxygen atom of the SO_2_ molecule and the distance that the latter has from the hydrogen atom of the functional group indicate the formation of a hydrogen bond, a bond that strongly enhances the binding energy. Furthermore, the gas molecule has a conformation that prompts the interaction of the electron-poor sulfur atom with regions of the functional group that have enhanced electrostatic potential.

The conformations of the optimized structures can be explained from the electron density redistribution plots that are depicted in Figure 3. Densities were plotted by using gOpenMol [22]. The electron density redistribution plots provide the means to predict the most thermodynamically favorable complexes between SO_2_ and organic molecules, as they depict the electrostatic interactions between the two monomers. Regions with high electron density, depicted in red, are primarily located around oxygen and nitrogen atoms, while regions of lower electron density, depicted in blue, are primarily located around hydrogen atoms. The electron density distribution in SO_2_ is such that the sulfur atom is enveloped by an electron-rich region that interacts with the electron-deficient regions present in the functionalized benzene molecule. Conversely, the electron-rich regions surrounding the oxygen atoms in SO_2_ interact with the electron-poor regions, specifically around the hydrogen atoms in the functionalized benzene molecules. The electron redistribution plots for all the dimmers are presented in Appendix A. It is clear that the complex with the highest binding energy, i.e., CO-C_6_H_5_PO_3_H_2_, also has the highest redistribution of electron density, whereas the least interacting dimer, i.e., CO-C_6_H_5_F, exhibits the lowest redistribution. The observed interaction between the two monomers is a typical Lewis acid–base interaction. 

Furthermore, a difference is observed between the initial SO_2_ angle (OŜO 119.1°) and the optimized angle obtained upon interaction with the modified benzene rings, as seen in Figure 4. It is evident that the geometry of SO_2_ changes as the interaction increases and the molecule’s angle extends to smaller values. Therefore, the binding energy and the aforementioned angle are inversely related. This observed correlation can be elucidated by the fact that in a sulfur dioxide molecule, the electronegativity of the two oxygen atoms is greater than that of the central sulfur atom. This difference is amplified even further with the interaction with another species, such as a ligand. Hence, the two oxygen atoms exert a greater pull on the bonded electron pair, thereby drawing it towards themselves and away from the central sulfur atom. Consequently, this results in a reduction of the electron pair repulsion between the bonded electron pairs on the central atom, leading to a decrease in bond angles between the groups.

After our initial investigation, our work proceeded to reveal how functionalization increases SO_2_ adsorption in MOFs. In this regard, we selected the top three performing functional groups to undertake the modification of IRMOF-8, which was the MOF of our choice. The last should meet certain prerequisites. First, we chose a MOF that belongs to the family of IRMOFs, which is a well-studied group. Moreover, IRMOF-8 has an appropriate linker that allows the accommodation of bulky functional groups without overlapping with the metal corner, and it is small enough to prevent the catenation effect. To predict the uptake isotherms, we employed Grand Canonical Monte Carlo simulations at a temperature of 298 K and a pressure range spanning from 0 to 3 bar.

The absolute volumetric uptake is presented in Figure 5. Given that the cell volume is constant throughout all examples analyzed, volumetric adsorption is proportional to the number of molecules adsorbed. The enhanced gas uptake of the modified IRMOF-8, prompted by the increased interaction between the functional groups and sulfur dioxide, is clearly observable. As demonstrated in Figure 5b, the enhanced performance of the modified structures is more pronounced at the low-loading region, specifically in the low-pressure range. We can see snapshots for the four studied cases taken from our GCMC simulations at 298 K and 0.01 bar in Figure 6, which are in perfect agreement with the calculated isotherms as the functionalized MOFs host substantially more SO_2_ gas molecules.

The enhancement of the modified materials is expected, as the adsorption process is primarily driven by the interaction energy at this particular pressure limit as opposed to other factors that may come into play at higher pressure ranges. Nevertheless, the uptake of the three modified IRMOF-8 appears to follow a different trend compared to the DFT calculations shown above. The SO_2_ uptake is greater in the modified –CNH_2_NOH functional group case, while this particular FG was found to have the third greatest binding energy compared to the other two cases. This is explained by the fact that the binding energy at the global minimum is just one of several factors that have a significant influence on the material’s adsorption capacity. The latter is greatly impacted by the size of the functional group and the different binding sites. The interaction energy of the –CNH_2_NOH functional group is 1 kcal/mol lower, but this is compensated by the greater size of the FG that creates more binding sites for the gas molecules.

These findings are further supported by Figure 7, where the isosteric heat of adsorption as a function of the SO_2_ loading is represented. The average binding energy of the gas molecules is greater in the case of the modified IRMOF8–CNH_2_NOH, while IRMOF8–PO_3_H_2_ and IRMOF8–OSO_3_H are second and third, respectively. As expected, the heat of adsorption for the parent material is the lowest. In addition, from Figure 7, it is evident that as the number of molecules grows, i.e., the most energetically favorable binding sites are occupied, the average binding energy of the gas molecules approaches the QM calculated values. This dictates that our QM data agree with the classical simulations.

Gravimetric uptake appears to follow the same trend at low loading but not at high pressure. As we can see in Figure 8, the order of the uptake curves changes at high pressure ranges where the unmodified material is performing slightly better. This is qualitatively expected since, in low pressure ranges, the strength of the interaction dominates the adsorption. Under high pressure conditions, where we have saturation, the pore volume and the framework weight play a critical role in the gravimetric uptake. In the upper limit, the higher mass of the modified MOFs—since they contain extra functional groups—together with the smaller pore size—due to the grafting of the pore by the functional groups—balances the winning due to the stronger interaction.

## 3. Methodology

We focused on optimizing the geometry and calculating the interaction energies between the benzene monomers and the gas molecule. First, we tested many different starting geometries for all optimizations. We used the Density Functional Theory (DFT) method and, more specifically, the Double Hybrid (DSD-BLYP) exchange correlation functional [23] and the D3(BJ) dispersion correction [24]. Resolution of Identity (RI) approximation is applied to our calculations to speed up the optimization process. In order to get reliable results in the DFT calculations, we had to choose a large orbital basis set, especially in terms of reference to the polarization function. Larger basis sets require more computational resources as they place fewer restrictions on electrons. However, they allow more accurate simulations of the exact chemical wavefunctions. Thus, we came to the choice of def2-TZVPP consisting of a triple zeta valence TZV and two sets of polarization functions. In addition, the appropriate auxiliary basis sets for the RI approximation were employed in the calculations. Binding Energies are calculated as the energy difference between the products and the reactants in the adsorption process, as defined in Equation (1):(1)$$BE=E_{system}-{(E}_{organiclinker}+E_{SO2})$$
where E*_system_* is the total interaction energy of the organic linker/adsorbates, while E*_organic_*_*linker*_ and E*_SO2_* are the total energy of the adsorbate-free organic linker structures and the SO_2_, respectively. All results were corrected for Basis Set Superposition Error (BSSE) using the counterpoise method [25]. All calculations were performed by Orca 4.2 [26,27].

Grand Canonical Monte Carlo simulations were used to compute the adsorption capacity of both the original and modified IRMOF-8 for sulfur dioxide. GCMC simulation is a powerful computational technique used in statistical mechanics to study the behavior of molecules or particles in a system where the number of particles is not fixed. It is based on the Metropolis algorithm, where the system undergoes a sequence of moves that change the number of particles in the system while keeping the chemical potential, temperature, and system volume constant. The chemical potential was determined using the Peng–Robinson equation of state and the two SO_2_ critical constants, Pc = 78.8 bar and Tc = 431 [28]. During each move, the energy change of the system is calculated and compared with a randomly generated number. If the energy change is less than or equal to the randomly generated number, the move is accepted, and the new state of the system is recorded. When describing the interactions between the IRMOF-8 and SO_2_ atoms, 6–12 Lennard-Jones (LJ) [29] and the Coulomb Potential were used, and each atom, whether a host or a guest, was treated definitely. The potential used has the following Equation (2):(2)$$V_{ij}=4\varepsilon_{ij}\left[{\left(\frac{\sigma_{ij}}{r_{ij}}\right)}^{12}-{\left(\frac{\sigma_{ij}}{r_{ij}}\right)}^{6}\right]+\frac{q_{i}q{}_{j}}{4\pi \varepsilon_{0}r_{ij}}$$
where *ε*_0_ is the vacuum permittivity constant; r_ij_ is the interatomic distance between interacting atoms i and j; q_i_ and q_j_ are the corresponding partial charges for atoms i and j; and e_ij_ and *σ*_ij_ are the LJ potential well depth and the repulsion distance between atoms i and j, respectively. A cut-off distance of 12.8 Å was applied to the Lenard-Jones potential, and Ewald summation was used for the long-range electrostatic interactions. The supercell used for the simulations was large enough so that each of its dimensions was at least two times the cut-off distance. Beyond this distance, the potential was shifted, and no tail corrections were applied. Both SO_2_ and the IRMOF–8 were considered to be rigid and represented by atomistic models. Sulfur dioxide was treated using the TraPPE model [28] as a three-center rigid molecule in the sense that neither the bond length nor the angle was allowed to vary during the simulations and instead kept constant at 1.432 Å and 119.3°, respectively. For the electrostatic interactions between SO_2_ molecules and the host material, point charges equal to q_o_ = −0.295 and q_s_ = +0.59 were placed at the oxygen and sulfur sites of SO_2_, respectively. The point charges of framework atoms were calculated using the CHELPG method: We first employed DFT calculations (RI-DSD-BLYP/def2TZVPP) on the isolated organic linker, and the metal corner with the CHELPG scheme applied to fit the point charges of the atoms. Subsequently, we modified the point charges to ensure the overall charge neutrality of the MOF unit cell. For the van der Waals interactions, we used potential parameters according to the TraPPE model developed by, i.e., *ε*/k_b_ = 73.8 K and *σ* = 3.39 Å for Sulfur atom and *ε*/k_b_ = 79.0 K and *σ* = 3.05 Å for the oxygen center. The essential potential parameter values for each MOF structure were derived from the Universal Force Field (UFF) [30]. The SO_2_–IRMOF-8 interactions were described using Lorenz–Berthelot mixing rules. [31,32].

For studying the SO_2_ adsorption in strategically functionalized MOFs with Grand Canonical Monte Carlo techniques, it is important to have an accurate classical potential to describe with great accuracy the interatomic interactions between the material and the gas molecule. Unfortunately, the potentials that are available in the literature [19,20,21] are not accurate enough and do not describe our system well. In this work, we modified the UFF classical potential in order to correctly reproduce ab initio results based on the following approach, which assesses the applicability of the parameters in question in great detail. Ab initio calculations were performed at the Density Functional Theory (DFT) level, and rigid scans of the distance of SO_2_ from the organic linker were carried out. During these calculations, we fixed the position of the functionalized benzene at the global (or local) minimum energy configuration of the dimer and sampled a selected distance, moving hydrogen from 6.0 to 1.5 Å toward the functional group position. Since the FGs contain many different atom types whereby parameters, *ε* and *σ*, play an important role in the potential energy surface, we also examined the local minima. For each different local minimum studied, a different FG atom plays the dominant role in the interaction energy. Finally, an in-home parametrization algorithm was used to adjust the parameters of the classical potential to accurately replicate the results obtained from these quantum chemical calculations. The fitting plots are shown in Appendix A.

During simulations, 50,000 cycles were used to allow the system to reach equilibrium and 100,000 more cycles to calculate the system properties. Each cycle consists of a number of steps equal to the number of guest molecules present in the system at that moment or equal to 20 if less than 20 molecules are adsorbed at that moment. All the GCMC simulations were conducted using the RASPA multipurpose code [33]. 

## 4. Conclusions

In conclusion, in this work, we studied the effect of linker functionalization in sulfur dioxide adsorption in MOFs. We calculated the interaction of SO_2_ with 41 strategically functionalized benzenes that can act as MOF linkers by employing quantum mechanics calculations. The -PO_3_H_2_ functional group has the greatest binding energy, which was calculated to be −10.1 kcal/mol at the RI-DSD-BLYP/def2-TZVPP level of theory. This constitutes a 2.5 times enhancement compared to the corresponding value for the unfunctionalized benzene (4.1 kcal/mol). All but five (-NCO, -NCS, -NC, -PH_2_, -F) functional groups favored a stronger interaction between the substituted benzene ring and the gas molecule. Using the redistribution electron density plots for the optimized dimers containing SO_2_ molecules, the nature of the interactions was further examined. It is observed that the higher the redistribution of the electron density, the higher the binding energy between the two monomers. This leads to the conclusion that the interaction is a typical Lewis acid–base interaction. Moreover, three of the top-performing functional groups (–PO_3_H_2_, –CNH_2_NOH, –OSO_3_H) were selected to modify the IRMOF-8 organic linker and calculate the SO_2_ adsorption capacity with Grand Canonical Monte Carlo simulations. The latter was employed at ambient temperature (298K) and pressure of up to 3 bar. Our results showed a significant increase in both the volumetric and gravimetric uptake, which is more profound at low pressure ranges where a ~350% increase is observed. This indicates that the suggested functionalization technique can be used to enhance the SO_2_ uptake capability not only in MOFs but in a variety of porous materials. We firmly believe that the results set out above can serve as a high-accuracy reference as well as lead synthetic efforts toward materials with high SO_2_ storage capacity and selectivity. 

## Figures and Tables

**Figure 1 molecules-28-03122-f001:**
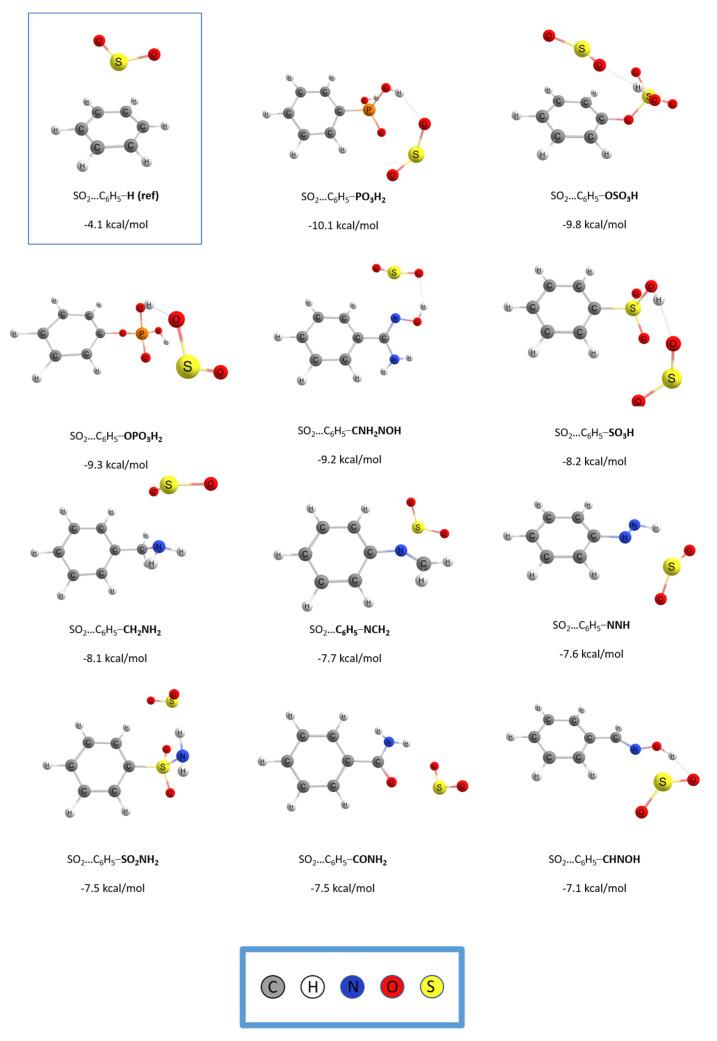
The optimized geometries for the 11 best performing functionalized benzene and the unfunctionalized for comparison, along with the corresponding binding energies calculated at the DSD–BLYP/def2–TZVPP level of theory, corrected for BSSE.

**Figure 2 molecules-28-03122-f002:**
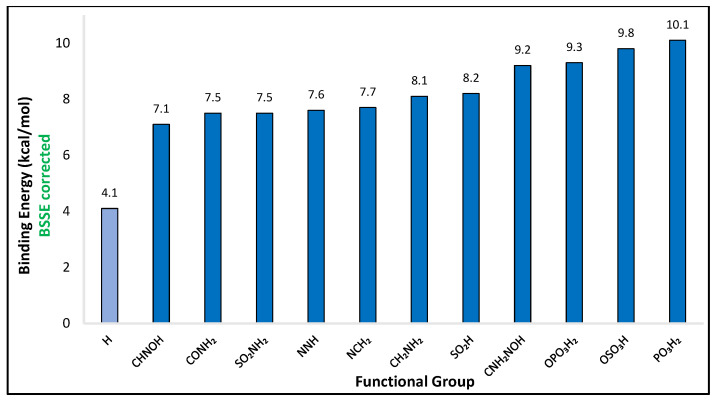
The binding energies of the 11 top-performing functionalized benzene together with the unfunctionalized one for comparison, calculated at the DSD–BLYP/def2–TZVPP level of theory, corrected for BSSE.

**Figure 3 molecules-28-03122-f003:**
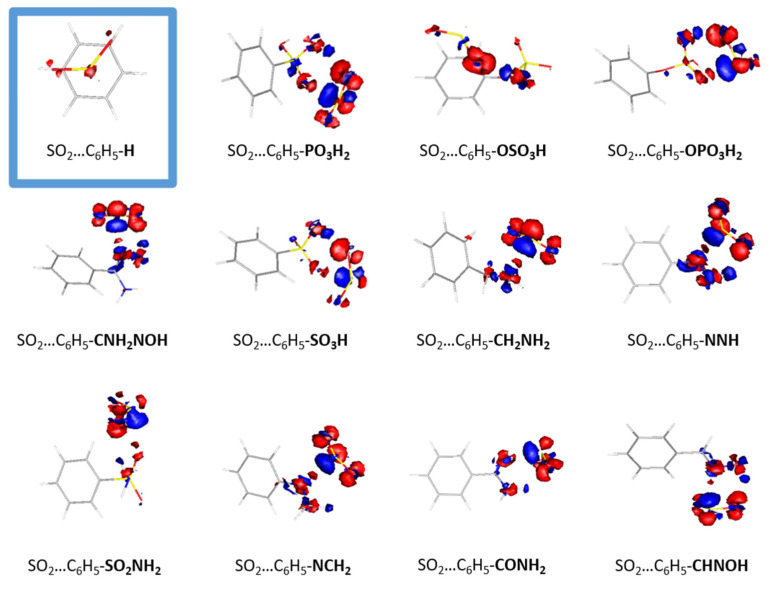
The electron density redistribution plots of the interacting molecules with SO_2._

**Figure 4 molecules-28-03122-f004:**
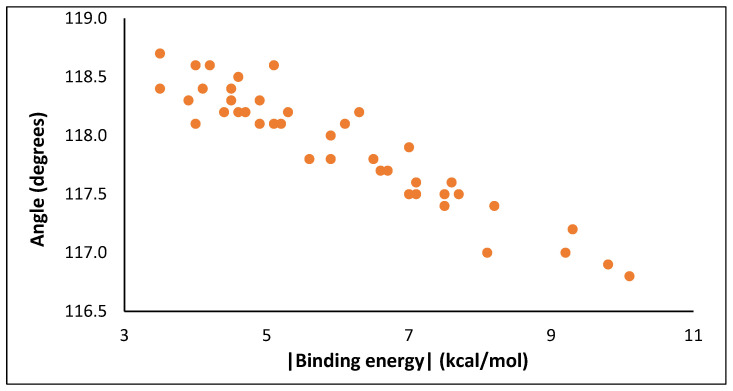
The angle (OŜO) of SO_2_ gas molecule as a function of the binding energy.

**Figure 5 molecules-28-03122-f005:**
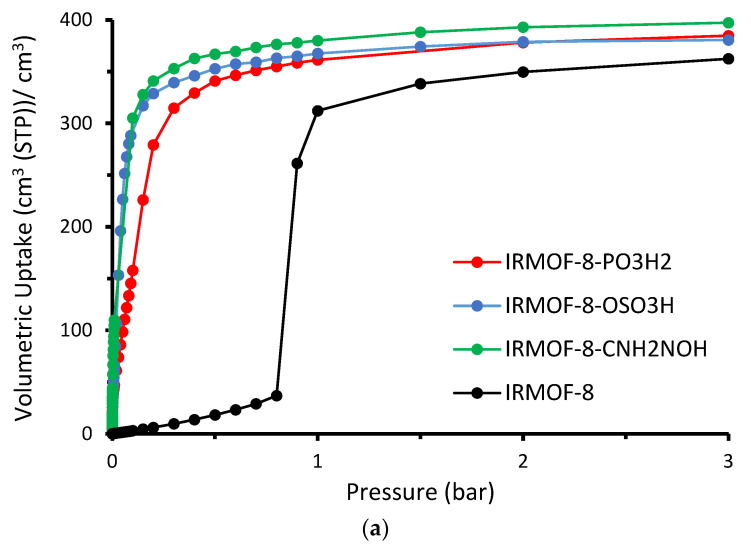
The absolute volumetric uptake isotherms for IRMOF–8 and IRMOF–8–n (n: –PO_3_H_2_, –OSO_3_H, and –CNH_2_NOH) at T = 298 K and pressure range from (**a**) 3 bar to (**b**) 0.01 bar.

**Figure 6 molecules-28-03122-f006:**
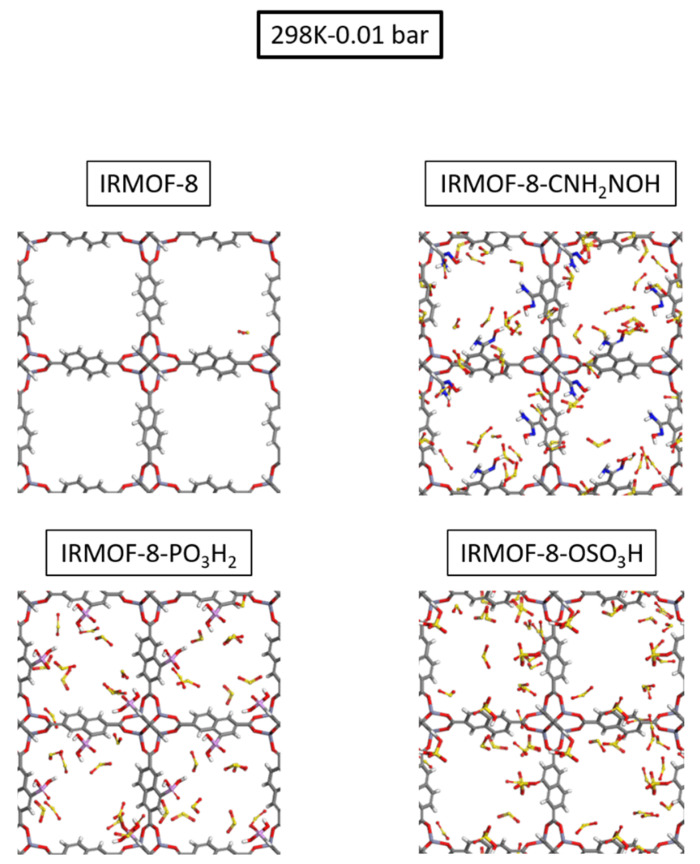
Snapshots of IRMOF–8, IRMOF–8–OSO_3_H, IRMOF–8–PO_3_H_2,_ and IRMOF–8–CNH_2_NOH taken from our GCMC simulations at 298 K and 0.01 bar.

**Figure 7 molecules-28-03122-f007:**
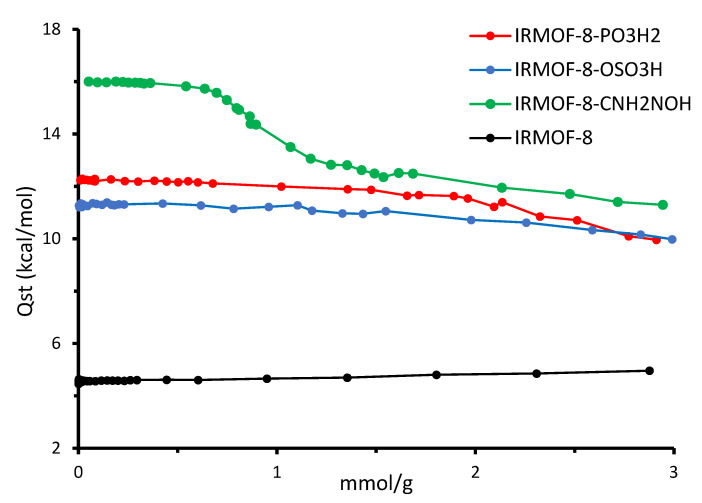
Isosteric heat of adsorption for the three modified IRMOF-8 materials and the parent one. Lines were added to guide the eye.

**Figure 8 molecules-28-03122-f008:**
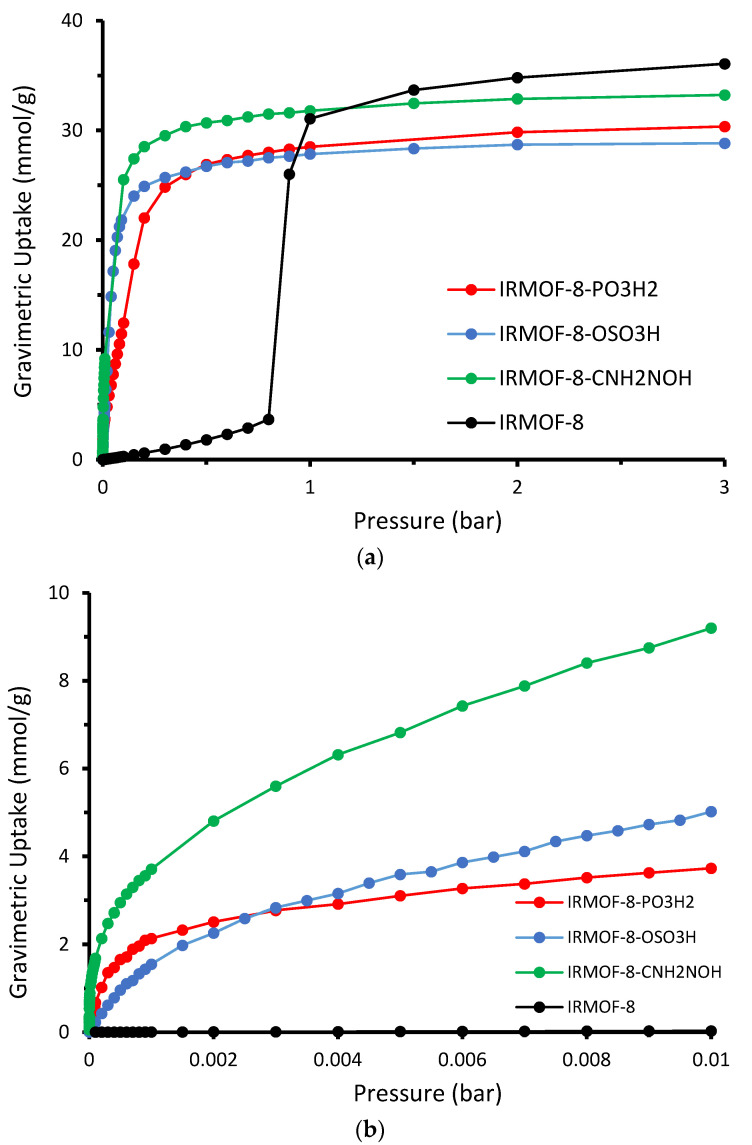
Absolute gravimetric isotherms for IRMOF–8 and IRMOF–8-n (n: –PO_3_H_2_, –OSO_3_H and –CNH_2_NOH) at T = 298 K and pressure range from (**a**) 3 bar to (**b**) 0.01 bar.

## Data Availability

The data presented in this study are available in Appendix A.

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
