# Peer review of "Multiscale Theoretical Study of Sulfur Dioxide (SO2) Adsorption in Metal–Organic Frameworks"

_molecules, 2023, doi:10.3390/molecules28073122_

Round 1

Reviewer 1 Report

In this manuscript, 41 functionalized groups were studied by the authors and their interactions with SO2 were calculated via DFT. The authors selected 3 top performance functional groups and decorated IRMOF-8 with them. Using modified force field, the SO2 adsorption capability of IRMOFs were evaluated using GCMC simulation. The work is suitable for the journal of Molecules after minor modifications.

1.      In the methodology part, the authors need to provide more details on the GCMC simulation, such as the equilibrium and initialization steps. The author should also mention how the framework’s charge is defined. The atomic charges for SO2, are equal to point charges qo = -0.295 and qs = + 0.59, are they obtained based on DFT calculation?

2.      For the possible anionic structures (-OH, -COOH, -SO3H) in DFT study, the authors should discuss whether to use a diffuse function for the def2-TZVPP basis set.

3.      As author mentioned that DFT-D3 method have been used for the dispersion correction in supporting information, it should be added in methodology part.

4.      Line 220, The author does not explain why the bond angle has become smaller, with the increasing of binding energy. Based on such result, flexibility of molecules should be considered in the GCMC simulation. Are the functional groups fixed in the DFT calculation? More details should be discussed.

5.      The resolution of Figure 2 is not enough for the readers to understand. The authors also need to mark the important position with atomic symbols in Figure 2, such as N, O and S.

6.      Figure 7, the Qst value of IRMOF-8-CNH2NOH is ~16 kcal/mol (~66.94 kJ/mol), such Qst is beyond the range of physical adsorption. The calculated interactions are based on the force field in this paper, why the Qst is so high?

7.      Typo mistake,

Line 62, “these material” should be “these materials”.

Line 76, “were” should be “where”.

Line 232, “he” should be “the”.

Reviewer 2 Report

The manuscript systematically investigated the SO2 adsorption of the designed MOFs material by using quantitative calculations as well as GCMC simulations. Few minor revisions are essential to consider this work for publication, the comments are as listed:

1. For the designed MOFs, the authors should add some explanation on why choosing IRMOF-8 for this study and the advantages of IRMOF-8.

2. GCMC modelling part of the manuscript needs more details. For instance, how many cycle steps were set? Please explain this in more detail.

3. Fig 1 lacks some information that which elements correspond to different colors.

4. Why the -PO3H2 group has the highest interaction energy with SO2, some expression could be added to explain this phenomenon.

5. The authors should include a brief review on materials that have been reported for SO2 adsorption and compare their performance with the proposed MOF.

Reviewer 3 Report

In this manuscript, the effect of linker functionalization in sulfur dioxide adsorption in MOFs is investigated using computer simulations. The SO2 adsorption in MOFs is an important topic. I have only minor comments for the authors to consider:  

1)  Since the authors invoke in sulfur dioxide adsorption in porous material, the Introduction should discuss and cite early papers on these topics, i.e. Liusheng Wang et al, Fluid Phase Equilibria. 567, 113710 (2023).

  2)  The authors may be interested in a paper by Zhou et al discussing  the effect of functionalization in noncovalent interactions.  See PCCP  21, 15310-15318 (2019); J. Chem. Phys. 148, 194106 (2018).  The authors have presented some very interesting computational data, and should focus on physical observables. 3) What are the critical constants for SO2 in the input file? Where did they take them? Any equation of state?

4) How did they model the polarizability for the SO2?

Round 2

Reviewer 1 Report

The author has revised the manuscript very well, and I think it is acceptable.

Reviewer 2 Report

The manuscript has been revised according to the reviewer's comments.